# The dietary effects of two strain probiotics (*Leuconostoc mesenteroides*, *Lactococcus lactis*) on growth performance, immune response and gut microbiota in Nile tilapia (*Oreochromis niloticus*)

**Assel Paritova[1], Akylbek Nurgaliyev[2], Gulbaram Nurgaliyeva[2], Nurzhan Abekeshev[2], Altynay Abuova[3], Faruza Zakirova[2], Grzegorz Zwierzchowski[4], Zhaxygali Kuanchaleyev[1], Saltanat Issabekova[1], Maigul Kizatova[5], Zaure Sayakova[6], Dinara Zhanabayeva[1], Yelena Kukhar[1], Ruslan Stozhkov[1], Botagoz Aitkozhina[1], Yevgeniy Mayer[1], Svetlana Bayantassova[2]\*, Angsar Satbek[1], Alexandr Andruchshak[1], Kaissar Kushaliyev[2]\***

**1** Saken Seifullin Kazakh Agrotechnical University, Astana, Kazakhstan, **2** Zhangir Khan West Kazakhstan Agrarian–Technical University, Uralsk, Kazakhstan, **3** International Engineering and Technological University, Almaty, Kazakhstan, **4** University of Warmia and Mazury, Olsztyn, Poland, **5** S.D. Asfendiyarov Kazakh National Medical University, Almaty, Kazakhstan, **6** Kazakh Scientific Veterinary Research Institute, Almaty, Kazakhstan

\* bayantasova65@mail.ru (SB); kaissark@mail.ru (KK)

## Abstract

The aquaculture industry has been growing rapidly over the past few decades to meet future animal protein demands. However, intensive aquaculture industry faces challenges such as growth abnormalities, high mortality rates, water quality and intestinal health deterioration. Administering probiotics can serve as a nutritional strategy to enhance the immune system and growth performance of fish influxes of gut microbiota. This study aimed to evaluate the impact of two dietary probiotic strains *L. mesenteroides* and *L. lactis* on the growth performance, immunity, and gut microbiota of Nile tilapia (*Oreochromis niloticus*). Fish were fed with basal and experimental diet supplemented by both *L. mesenteroides* and *L. lactis* bacteria at $10^6$ cell/g for 8 weeks. Feeding a combination of *L. mesenteroides* and *L. lactis* resulted in significant improvements in feed utilization parameters (PER and FER) (P < 0.001), alternative complement pathway activity, intestinal lactic acid bacteria count (P < 0.012), mucus secretion (P < 0.002) and peroxidase activity (P < 0.001) compared to the control groups. Serum lysozyme activity also exhibited a significant increase in the *L. mesenteroides* and *L. lactis* dietary group (P < 0.011) compared to the control and single probiotic supplemented diet groups. Furthermore, Nile tilapia fed the *L. mesenteroides* and *L. lactis* supplemented diet showed enhanced growth performance metrics (weight gain, final weight and specific growth rate) compared to those fed control and single probiotic supplemented diets (P < 0.022). Additionally, superoxide dismutase activity was significantly elevated in the *L.mesenteroides* and *L. lactis* supplemented diet groups compared to the control and single *L.mesenteroides* supplemented diet groups (P < 0.017). These findings

**Data Availability Statement:** The original data presented in the study are openly available in FigShare at 10.6084/m9.figshare.27050416.

**Funding:** This research was funded by the Science Committee of the Ministry of Science and Higher Education of the Republic of Kazakhstan (grant no. AP19576848) The funders had no role in the design of the study; in the collection, analyses, or interpretation of data; in the writing of the manuscript, or in the decision to publish the results.

**Competing interests:** The authors have declared that no competing interests exist.

strongly indicate that a dietary combination of *L. mesenteroides* and *L. lactis* probiotics could function as a beneficial immunostimulant feed supplement in Nile tilapia aquaculture.

## Introduction

Over the past few decades, there has been considerable growth in the aquaculture industry, positioning it to effectively address the future animal protein demands of the human population [1]. Tilapia ranks among the most extensively farmed fish globally, representing one of the most widely practiced forms of aquaculture. They are highly suitable for aquaculture due to their ease of cultivation, rapid growth rates, and relatively high tolerance to environmental stressors compared to other fish species [2]. Nonetheless, intensive aquaculture encounters challenges that have raised concerns regarding fish health, encompassing growth abnormalities, mortality rates, and disease outbreaks [3]. Consequently, farmers are forced to use prophylactic antibiotics to mitigate these challenges [4,5].

Antibiotics are commonly employed in aquaculture to manage diseases and enhance growth performance. However, their usage in this context has the potential to disrupt intestinal microbiota and foster the emergence of antibiotic-resistant bacteria, posing risks to aquatic life [5,6]. Consequently, previous studies aimed to devise alternative feed additives to substitute antibiotics in aquaculture practices [7–10]. The application of probiotics as dietary additives has been recently proposed as a viable alternative to antibiotics [11,12]. Their application aims to reduce the colonization gastrointestinal mucosa by pathogenic population, thereby enhancing the health of mammals and fish [13–15]. Additionally, lactic acid bacteria (LAB) were ability to increase growth performance and survival rates, prevent intestinal disorders such as diarrhea, neutralize antinutritional factors [16,17]. Moreover, advantageous intestinal microbiota can produce enzymes that have the potential to enhance the digestibility and uptake of nutritional components [18]. It was documented that incorporating a combination of probiotics into the diet of *Labeo rohita* fingerlings resulted in improved nutrient digestibility, enhanced growth, a higher protein efficiency ratio, and a reduced feed conversion ratio compared to the use of individual strains [19]. In this study, we have used two probiotics, L. mesenteroides and L. lactis, included in the diet of Nile tilapia (*Oreochromis niloticus*). These LAB were isolated from the gastrointestinal tracts of different fish species (e.g. northern pike, zebrafish, carp,). They are important in aquaculture as they help balance the intestinal microbiota, enhance growth performance, improve digestion and nutrient absorption, and boost the immune response

This study examined the effect of two probiotic strains in fresh water fish Nile tilapia on growth performance, cultivable gut bacteria, and innate immune response, as well as body weight gain.

## Materials and methods

### Animal ethics

The experiment was carried out according to the European Convention for the Protection of Vertebrate Animals used for Experimental and Other Scientific Purposes guidelines [20]. The protocol was approved by the Committee on the Ethics of Animal Experiments of the Saken Seifullin Kazakh Agrotechnical University (permit number: 2).

## Experimental design

Overall, 395 healthy juvenile Nile tilapia with body weight averaging 45±2g were purchased from local commercial fish farm. Fish were then equally divided in four groups in 500L tank: control group without probiotic supplementation (C) 96 heads and three experimental groups *L. mesenteroides (LM), L. lactis (LL) and a combination of both (mix)*–each consisting of 96 fish per group. Subsequently, the fish were randomly distributed into twelve 100-liter tanks, with thirty-two fish per tank and two replicate tanks per treatment. Each tank was fitted with both an inlet and an outlet and maintained in a flow-through freshwater system. Prior experiment onset, fish allowed to acclimatize in 500L tank with 5mg/L $O_2$ aerated water for 14 days. Water quality parameters were assessed by employing a thermometer to measure temperature and an oxygen meter (Model FEBO 330, EPF Instruments, Fisher Scientific, USA) to measure dissolved oxygen levels. Throughout the experiment, water temperature was maintained at 27˚C. The pH level was monitored with a portable pH meter (Fisherbrand AP 115). During the acclimatization period fish were fed a control diet without probiotic supplements *ad libitum* twice a day at 9:00 a.m. and 17:00 p.m. After completing the feeding trial (8 weeks), all experimental fish subjected a 24-h fasting. Following this, survival rate, individual body weight, immune parameters, and gut microbiota composition of fish from each tank were determined.

## Isolation of probiotics

*L. mesenteroides* and *L. lactis* LAB starins were isolated from intestines of common carp (*Cyprinus carpio*) fingerlings following protocol described by [3]. In brief, a 50 μl sample of intestinal microflora from fish was dissolved in 500 μl distilled water (1:10 ratio). Subsequently, using a disposable pipette, 100 μl of the suspension was introduced to the MRS agar medium and incubated at 37˚C for 48-h [21]. Glass beads were utilized to ensure uniform growth of isolated colonies. Eight to twelve glass beads (pre-sterilized) were placed into the Petri dish containing the inoculum. The probiotics obtained were washed three times in distilled water and plates were sent for genomic confirmation.

## PCR

For DNA extraction, GeneJET PCR Purification Kit (Thermo Fisher Scientific, USA) was used according to the manufacturer's instructions. Following reference strains were utilized as a positive controls: National Collection of Type Culture (NCTC) AB023246 for *L. mesenteroides* and (NCTC) M58837 for *L. lactis*. For genotyping of isolates, multiplex PCR (mPCR) was run according to the protocol described by Ali, Kot [22]. The amplification of 16S mDNA fragments in a Thermal Cycler (Eppendorf Mastercycler) were as follows: 95˚C for 2 min 30 s, 95˚C for 1 min, 55˚C for 1 min, 72˚C for 1 min 20 s (35 cycles), and 72˚C for 2 min. The PCR products obtained were administered in 1% agarose gel electrophoresis in the presence of Ethidium bromide, and the results were visualized using Bio-Imaging Systems (MiniBIS Pro, Israel).

## Preparation of experimental diets

Commercial feed (F.lli Fragola, S.P.A, Italy) without probiotic supplements was used as a basal control diets. Basal diet composed of fishmeal, soybean meal, milled wheat, extruded peas, potato starch, vegetable oil, tricalcium phosphate, premix, yeast (Table 1).

The culture medium was prepared using a water-based solution with the addition of up to 1.5% glucose. Activated strains of *L. lactis* and *L. mesenteroides* were used as inoculants, added at a concentration of 0.05%. Briefly, in distilled water, the following components were added:

**Table 1. Composition and nutrient levels of the basal diet (g kg-1).**

| Ingredients | Diet |
|---|---|
| Fishmeal | 30 |
| Soybean meal | 42 |
| Milled wheat | 7 |
| Extruded peas | 8 |
| Potato starch | 6.7 |
| Vegetable oil | 4 |
| Vitamin/Mineral mixture | 1.1 |
| Premix | 0.1 |
| Yeast | 0.05 |
| Gelatin | 1 |
| Proximate composition | |
| Moisture | 8.79 |
| Crude protein | 35.8 |
| Crude lipid | 7.39 |
| Crude ash | 6.44 |

1.5% glucose, 0.1% buffer salts, 0.1% ascorbic acid, 5% peptone, 1.5% microbiological agar, and 0.5% yeast. The pH of the medium was adjusted to (7.0±0.1). The prepared medium was sterilized at 121˚C for 30 minutes, then cooled to (37±1˚C. Then, an inoculum containing the activated strains *L. lactis* and *L. mesenteroides* at a concentration of 0.05% was introduced. Biomass cultivation was carried out under batch fermentation for (24±2) hours, with pH neutralization using $Na_2CO_3$ at the 12-hour mark. After cultivation, the upper layer of the culture broth was separated, and the bacterial suspension was cooled to (4±2˚C, dispensed into aseptic vials (10–12 ml), sealed, and labeled. The experimental diet was prepared by introducing $1 \times 10^7$ CFU/g of a single *L. mesenteroides* or/and *L. lactis* and mixed probiotic in the basal diet. To do so, these LAB were incorporated into the basal diet using the spraying method while stirring. Then, 165g of gelatin was dissolved in 500 ml of distilled water to create a waterproof membrane for the feed. Next, the dietary components underwent grinding into a fine powder, followed by thorough agitation and subsequent cold extrusion into pellets 0.5 mm in diameter. Finally, the diluted gelatin was sprayed onto LAB incorporated basal diet pellets under thorough stirring. The complete diet were then air-dried overnight at room temperature and stored in plastic containers at −4˚C until required.

## Growth performance

The survival rate was calculated according to following equation: Survival (%) = 100 × (final number of fish/initial number of fish). The calculation of weight gain (WG), feed conversion ratio (FCR), protein efficiency ratio (PER) and specific growth rate (SGR) were performed according to protocol described by Allameh, Yusoff [23]. In brief, WG in grams is calculated by subtracting the initial weight from the final weight. SGR is determined by the formula: SGR = 100 (ln W2—ln W1) / T, where W1 and W2 represent the initial and final weights, respectively, and T signify the number of days during the feeding period. SGR indicates the percentage of growth per day for each fish.FCR is computed as the ratio of feed intake in grams to weight gain in grams, while PER is determined by the ratio of weight gain in grams to protein intake in grams.

## Determination of intestinal bacteria activities

To determine the total intestinal microbiota and LAB count, eleven fish were sampled from each study group following 24-h feeding cessation. Prior experiment onset, fish were anesthetized using tricaine methanesulphonate (MS 222) and then transferred on to sterile stainless desk. Before opening, fish carcasses were rinsed with 70% ethanol to eliminate surface contamination.

The whole intestine was extracted, and 1 cm sections from both the anterior and posterior ends were discarded. Then the whole intestinal tract was dissected, and rinsed three times in PBS. Subsequently, tenfold serial dilution was run in PBS. 100 ml aliquots of the diluted solution were then plated onto duplicate Trypto-Soya agar (TSA) media to determine total bacterial counts. To isolate viable LAB species, 100 ml of each dilution was introduced in MRS (DeMan, Rogosa, and Sharpe) agar and allowed to incubate at 25°C for 3 to 5 days. Colony-forming units of viable bacterial populations were enumerated according to protocol reported by Ramos, Batista [24].

## Immune responses

Alternative complement pathway activity (ACH50) was determined according to protocol of Yano [25]. In brief, 0.1ml aliquots of diluted serum were transferred in to plastic tubes and adjusted to a total volume of 0.25 ml using barbitone buffer. Rabbit red blood cells (RaRBC) were then added to each tube in a volume of 0.1 ml and allowed to incubate for 1-h at 20°C. Next, after incubation, 3.15 ml of 0.9% NaCl solution was introduced and samples were centrifugated at 1600 x g for 10 min at 4°C to remove RaRBC derbies. Finally, obtained supernatant was measured for optical density at 414 nm, and a lysis curve was generated. The volume resulting in 50% hemolysis was identified and utilized to assess the complement activity of the sample.

The bactericidal activity of serum and mucus was assessed following the method described by [26]. To do so, samples were diluted in 1:1 ratio with Tris buffer adjusted to pH 7.5. Then, diluted mixture was agitated with pre-prepared 0.001 mg/ml *Escherichia coli*, IAM1239 bacterial suspension (RIBSP, Kazakhstan) and allowed to incubate overnight at room temperature. The standard plate counting method was employed to calculate colony-forming units (CFU).

To measure lysozyme activity in serum and mucus samples a turbidimetric assay was utilized. Briefly, 10 μl of samples were mixed with 190 μl of *Micrococcus lysodeikticus* bacterium (0.2 mg/PBS adjusted to pH 7.4) in 96 flat-bottomed well plates, and the mixture was allowed to incubate for 5 min at room temperature. Absorbance was measured at 450 nm using Biotec 800 TS (Agilent Technologies, USA). Lysosome activity was determined based on the amount of sample resulting in a decrease in optical density of 0.001 min$^{-1}$.

The total peroxidase content in serum was assessed according to method described by Salinas, Abelli [27]. To do this, 15 μl of serum and 35 μl of Hank's Buffered Salt Solution were added to 96 flat-bottomed well plates. Subsequently, 50 μl of 3,3′,5,5′-Tetramethylbenzidine (TMB, Thermo Scientific, USA) was added to each well. The diluted serum was then incubated for 20 min, and the color-developing reaction was halted by introducing 50 μl of 2 M sulfuric acid. The optical density was measured at a wavelength of 450 nm using a plate reader.

Serum superoxide dismutase (SOD) activity was quantified by the percentage inhibition rate of the enzyme's reaction using Ransod kit (Randox, Crumlin, UK) following the protocol described by Sun, Yang [28]. The specific SOD activity units were adjusted by milligrams of protein (mg protein)$^{-1}$. One unit of SOD is determined as the quantity of the enzyme present in 20 ml of the sample solution, which inhibits the reduction reaction of nitroblue tetrazolium (NBT) by 50% at wavelength of 550 nm.

## Statistical analysis

Graphs were compiled and statistical analyses were conducted to protocol described by Bayan-tassova, Kushaliyev [29,30] using Prism 9 software (GraphPad Software, Inc., San Diego, CA). The data were subjected to Kolmogorov–Smirnov and Levene's tests to assess the normality of the distribution and verify the homogeneity of variances. Data were analysed by one-way ANOVA with Tukey's post hoc test. Statistical significance between groups was denoted when $P < 0.05$ using asterisks, as detailed in the respective figures.

## Results

### Demonstration of growth performance, nutrient utilization and survival

The growth parameters, feed uptake and survival of control and fish fed *L. mesenteroides* or/ and *L. lactis* diet are presented in Table 2. Fish fed the mixture of *L. mesenteroides* and *L. lactis* diet demonstrated a substantial increase in WG, FBW, and SGR ($P < 0.033$) compared to the groups fed single *L. mesenteroides* or the control diets. Additionally, the PER and FER profiles were significantly higher in fish fed on mixture of *L. mesenteroides* and *L. lactis* diet ($P < 0.001$), whereas these profiles did not differ between single probiotic supplemented groups ($P > 0.991$) Moreover, FER profile did not differ significantly between fish fed single probiotic supplemented diets and those fed control diet. Furthermore, a significant increase in the protein gain (PG) profile was observed in fish fed *L. mesenteroides* or/and *L. lactis*-containing diet compared to the control group ($P < 0.021$). Feed intake profile was enhanced by both *L. mesenteroides* and *L. lactis* dietary treatments, with fish receiving only *L. lactis* exhibiting significantly greater feed intake compared to the control group ($P < 0.041$). However, no significant difference was observed between fish fed single probiotic supplemented diets ($P > 0.087$). Survival rates were high across all treatment groups, and no mortality was observed.

**Table 2. Growth performance of Nile tilapia fed with *L. mesenteroides* or/and *L. lactis* supplemented diets for 8 weeks.**

|  | Diets | | | |
|---|---|---|---|---|
|  | **Control** | ***L. mesenteroides*** | ***L. lactis* probiotics** | **Mixture of *L. mesenteroides* and *L. lactis* probiotics** |
| [1]IBW (g) | 46.3±1.32 | 45.4±3.44 | 45.9±1.02 | 45.8±2.11 |
| [2]FBW (g) | 68.4±3.26[a] | 79.2±1.16[b] | 78.5±2.15[b] | 92.4±4.54[c] |
| [3]WG (%) | 22.4±3.01[a] | 31.7±1.52[b] | 31.2±1.74[b] | 47.3±2.02[c] |
| [4]SGR (%) | 1.96±0.11[a] | 2.32±0.5[b] | 2.30±0.3[b] | 2.82±0.2[c] |
| [5]FI (g) | 12.42±0.47[a] | 12.88±1.05[ab] | 13.01±0.6[b] | 13.24±0.8[b] |
| [6]FER | 2.33±0.2[a] | 2.64±0.8[ab] | 2.60±1.12[ab] | 2.97±0.5[c] |
| [7]PER | 3.05±0.4[a] | 3.44±0.18[b] | 3.52±0.7[b] | 3.65±0.03[c] |
| [8]PG | 141.23±7.2[a] | 158.42±2.13[b] | 162.32±3.04[b] | 179.31±4.05 |

[1]IBW—initial body weight

[2]FBW—final body weight

[3]WG—weight gain

[4]SGR—specific growth rate

[5]FI–feed intake

[6]FER—feed efficiency ratio

[7]PER—protein efficiency ratio

[8]PG—protein gain. Values represent means ± SEM of triplicate groups. Different superscripts in the line represent significant difference $p \leq 0.05$.

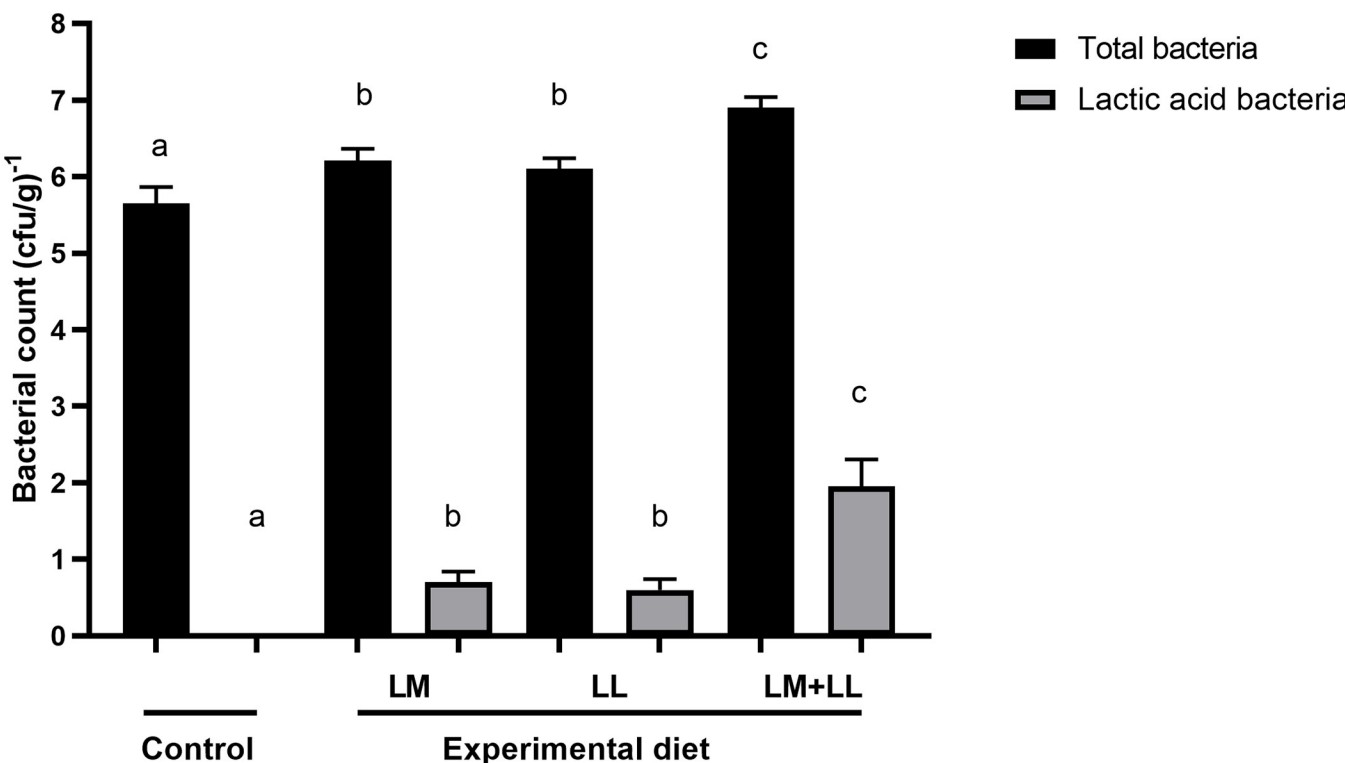

**Fig 1. Comparison of total bacteria and LAB content of Nile tilapia intestines fed with control and probiotics supplemented diets for 8 weeks.** Data presented as the mean ±SEM from two combined independent experiments (n = 10 fish/group). Bars with different alphabet are statistically different (P ≤ 0.05). Bars with the same alphabet are statistically insignificant different (P ≥ 0.05). * LM; LL–*L. mesenteroides*; *L. lactis*.

### Microbiological analyses of gut microbiota

Both the control and experimental groups showed no detectable LAB bacteria in the intestines before the experiment. Fish fed with single and the mixture of LAB-supplemented diets demonstrated significantly greater levels of the total intestinal microbiota compared to the control group. (P < 0.012) (Fig 1). Moreover, fish fed with the mixture of LAB-supplemented diets exhibited significantly higher cultivable LAB compared to the fish groups fed with single diets (P < 0.027) (Fig 1).

### Demonstration of immune responses

The rate of ACP were significantly elevated in fish fed *L. mesenteroides* or/and *L. lactis* supplemented diets than that of control group (Fig 2). Likewise, serum and mucus bactericidal activity (Fig 3), mucus lysozyme activity (Fig 4), peroxidase activity (Fig 5) demonstrated a considerable elevation in fish fed with single or mixture of probiotics supplemented diets compared to control group. Similarly, superoxide dismutase inhibition rate was exhibited significantly greater in fish groups fed with single *L. lactis* and mixture diets compared to other groups (Fig 6). Moreover, peroxidase activity (Fig 5) and ACP levels (Fig 2) in fish fed the combined L. mesenteroides and L. lactis diet were significantly elevated compared to those in fish fed single probiotic supplemented diets.

Additionally, when compared with the control and single probiotic supplemented diet groups, lysozyme activities of serum was significantly higher in fish fed mixture of *L. mesenteroides* and *L. lactis* diets (P < 0.011; Fig 4). Furthermore, the amounts of secreted mucus was

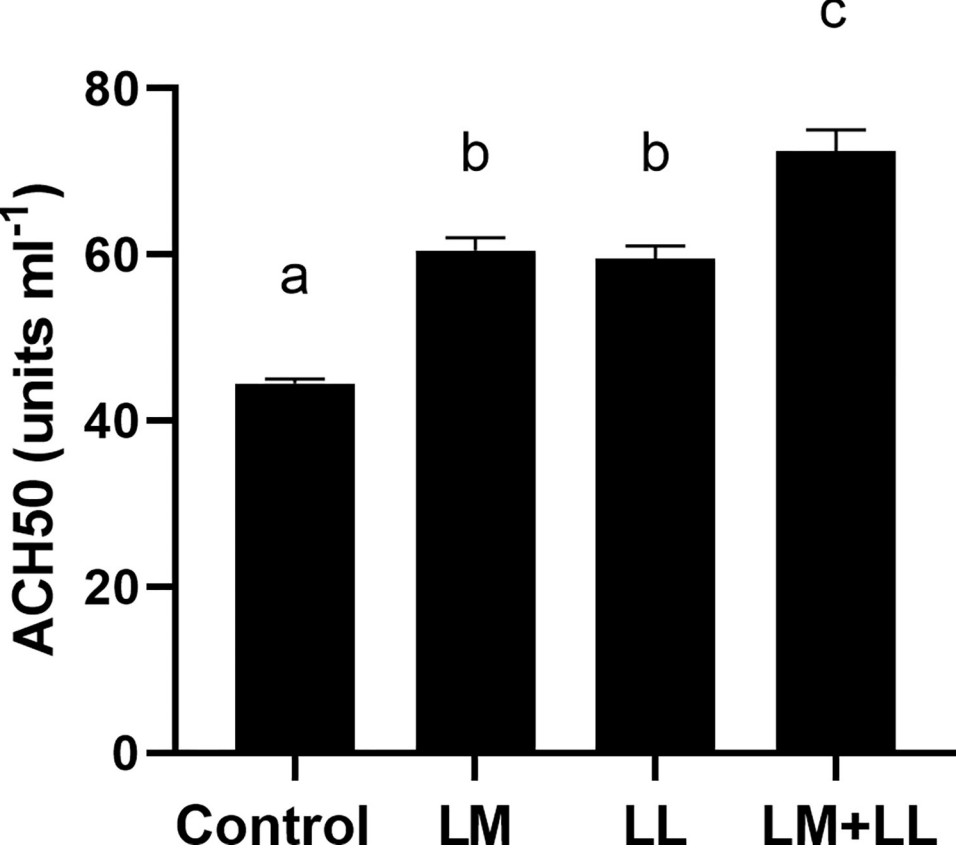

**Fig 2. Alternative complement pathway of Nile tilapia fed with control and probiotics supplemented diets.** Data presented as the mean ±SEM from two combined independent experiments (n = 10 fish/group). Bars with different alphabet are statistically different (P ≤ 0.05). Bars with the same alphabet are statistically insignificant different (P ≥ 0.05). * LM; LL–*L. mesenteroides*; *L. lactis*.

significantly higher in fish groups supplemented with *L. mesenteroides* or/and *L. lactis* diets compared to control group (P < 0.002; Fig 7).

## Discussion

An increasing body of research has illustrated the application of single lactic acid bacteria species as a probiotic treatment to enhance growth performance, immune response, feed utilization and gut microbiota modulation in fish [31–33]. However, there has been limited focus on exploring the potential effects that may arise from the concurrent oral administration of multiple LAB species to fish [3,34–36]. Hence, this study was carried out to assess the dietary effect of *L. mesenteroides* and *L. lactis* on the growth performance, immunity and gut microbiota of Nile tilapia.

The findings of this study suggest that administering both *L. mesenteroides* and *L. lactis* into supplementation diet significantly enhances the growth performance of Nile tilapia compared to the control group over a 8 weeks period. This outcome is consistent with previous research demonstrating that probiotic supplementation, particularly with *Lactobacillus* species, improves the growth performance of various fish species, including olive flounder (*P. olivaceus*) [35], Nile tilapia (*Oreochromis sp*) [21], gilthead sea bream (*Sparus aurata*) [37]. Furthermore, *L. lactis* has been found to enhance the growth performance of olive flounder [38] and

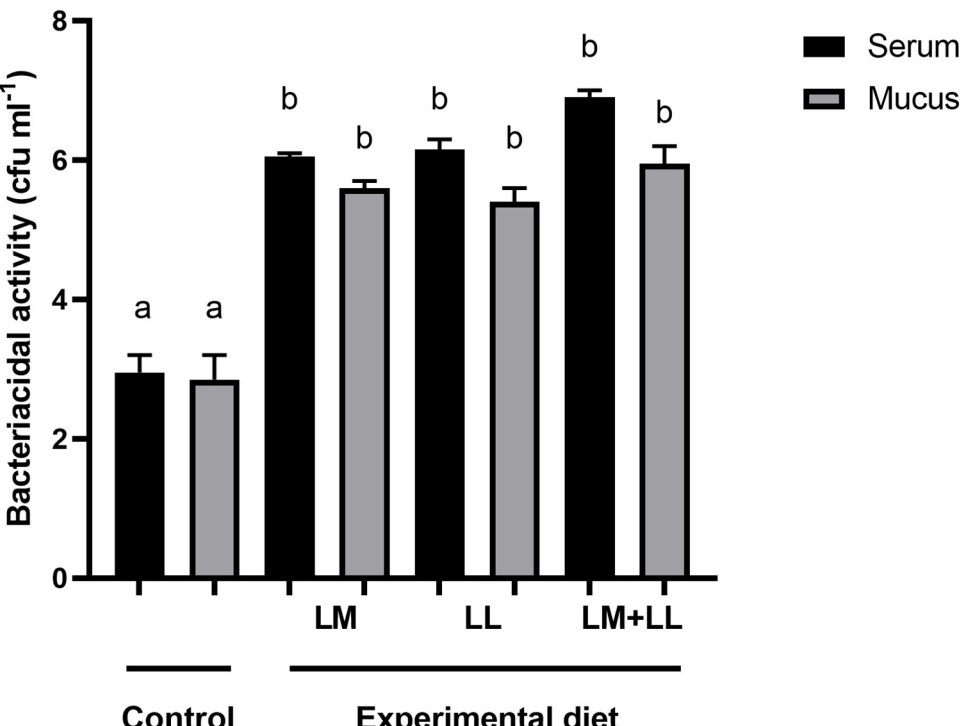

**Fig 3. Serum and mucus bactericidal activity of Nile tilapia fed with control and probiotics supplemented diets.**
Data presented as the mean ±SEM from two combined independent experiments (n = 10 fish/group). Bars with different alphabet are statistically different (P ≤ 0.05). Bars with the same alphabet are statistically insignificant different (P ≥ 0.05). * LM; LL—*L. mesenteroides*; *L. lactis*.

grouper (*Epinephelus coioides*) [39]. Additionally, the results obtained in this study showed that fish fed a mixture of *L. mesenteroides* and *L. lactis* exhibited higher final weight, weight gain, and specific growth rate compared to the groups fed the control and single probiotic diets. Similarly, Allameh, Yusoff [23] reported that administering multistrain probiotics, particularly *L. mesenteroides*, enhanced the growth performance of Javanese carp (*Puntius gonionotus*). In the current study, fish fed the diet supplemented with a mixture of *L. mesenteroides* and *L. lactis* showed significant improvements in feed intake and feed utilization parameters. This observation suggests that the probiotic supplement may have influenced the palatability of the feed, consequently resulting in slight increases in the growth rate and feed intake of fish. To the best of our knowledge, this study represents the first dietary incorporation of *L. mesenteroides* and *L. lactis* leading to enhanced growth and feed utilization in Nile tilapia. Xia, Cao [40] proposed that probiotics primarily influence Nile tilapia by augmenting host nutrition, thereby stimulating digestive enzymes and ultimately leading to enhanced growth and feed efficiency ratios. Moreover, the colonization of probiotic microorganisms in the intestine enhances microbial equilibrium, subsequently improving feed utilization and nutrient absorption [41–43].

Improvements in growth performance and feed utilization may also be attributed to enhanced intestinal digestive functions in fish owing to probiotic supplementation (*L. mesenteroides and L. lactis*), which likely includes increased activities of digestive enzymes. Previous studies have shown that supplementation with *Lactobacillus* species leads to improved nutrient digestibility [9]. It is well-established that the activity of digestive enzymes is positively correlated with the digestive capacity of fish [44], thereby enhancing their ability to extract nutrients

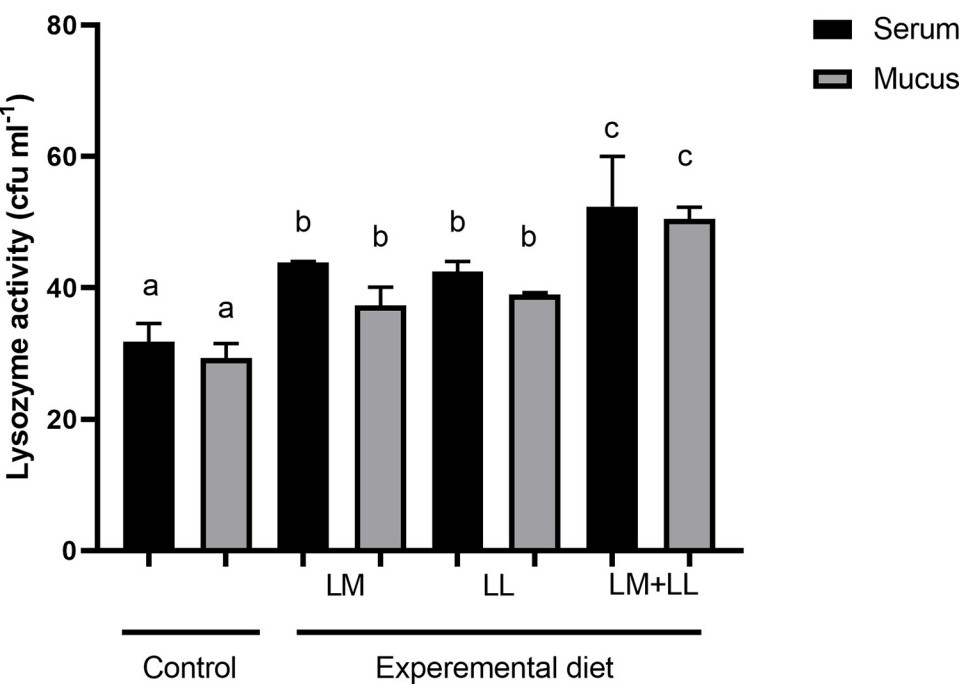

**Fig 4. Serum and mucus lysozyme activity of Nile tilapia fed with control and probiotics supplemented diets.** Data presented as the mean ±SEM from two combined independent experiments (n = 10 fish/group). Bars with different alphabet are statistically different (P ≤ 0.05). Bars with the same alphabet are statistically insignificant different (P ≥ 0.05). * LM; LL—*L. mesenteroides*; *L. lactis*.

from food [45]. It has been reported that digestive organs are highly sensitive to food composition, leading to immediate changes in the activities of digestive enzymes ultimately affecting fish growth and health [46]., Furthermore, bacteria secrete proteases to cleave peptide bonds in proteins, breaking them down into free amino acids and constituent monomers, thereby improving the animal's nutritional status. Bacterial enzymatic hydrolysis has been demonstrated to improve the bioavailability of lipids, protein and dry matter [47], potentially leading to the increased growth and nutrient utilization observed in this study.

The current study revealed a significant enhancement in both total intestinal bacteria and lactic acid bacteria counts, further supporting the improvements observed in growth and feed utilization. Similar to the findings of Mohapatra, Chakraborty [19], a substantial increase in total bacterial counts was noted when fish were fed diets supplemented with probiotics. This increase could potentially lead to improved fish health and immunity, thereby promoting growth. It has been documented that the colonization rate of bacteria in the digestive tract is directly influenced by the level of dietary bacteria intake [48]. In our investigation, the elevated adherence of microorganisms introduced via dietary supplementation could explain the improved nutrient utilization and growth observed in the fish. Numerous studies have suggested that probiotics function by colonizing the host and secreting various growth-promoting compounds [49,50].

Previous studies have documented the crucial interplay between intestinal epithelial cells and probiotics in regulating the mucosal immunity of the fish gut. This interaction modulates both the physical and immunological barrier functions of the intestine and stimulates an immune response [51–55]. The results of the current investigation also indicate that the dietary combination of both *L. mesenteroides* and *L. lactis* led to an enhanced immune response. These improvements included increased lysozyme and bactericidal activities in both serum

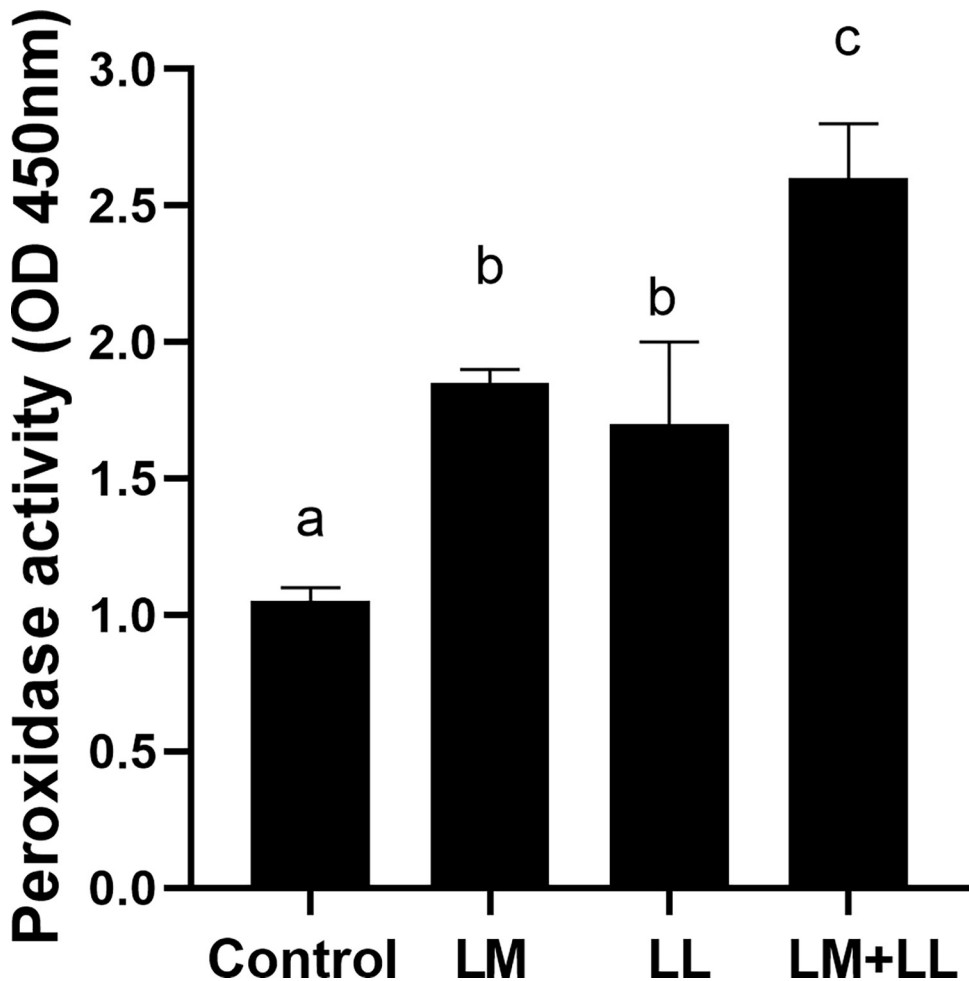

**Fig 5. Serum peroxidase activity of Nile tilapia fed with control and probiotics supplemented diets.** Data presented as the mean ±SEM from two combined independent experiments (n = 10 fish/group). Bars with different alphabet are statistically different (P ≤ 0.05). Bars with the same alphabet are statistically insignificant different (P ≥ 0.05). * LM; LL —*L. mesenteroides*; *L. lactis*.

and mucus, as well as enhancements in the alternative complement pathway, superoxide dismutase, and peroxidase activities. Likewise, the beneficial effect of combined probiotics in the diet have been documented to trigger innate immunity in rainbow trout (*Oncorhynchus mykiss*) [56], *Labeo rohita* [11], olive flounder [35]. Others reported that combined utilization of bacterial strains with complementary properties within the gut environment can enhance or extend the favorable health-promoting effects and immune response of the host [57]. Several authors have reported that external factors such as probiotics, vitamins, and nutrients influence the activity of the alternative complement pathway in fish [58,59]. In the present study, the activity of the alternative complement pathway emerged as one of the immune system parameters most affected, showing a significant increase in all of the experimental groups. These findings are similar to those described by Giri, Sukumaran [34] and Sun, Yang [39].

It was reported that bactericidal activity is a critical component of the host's defence mechanisms against pathogens [9,60]. The outcomes obtained in our study indicate that the greatest levels of serum and mucus bactericidal activity were observed in fish receiving diets supplemented with *L. mesenteroides* or/and *L. lactis*. The lysozyme activities in fish have been shown

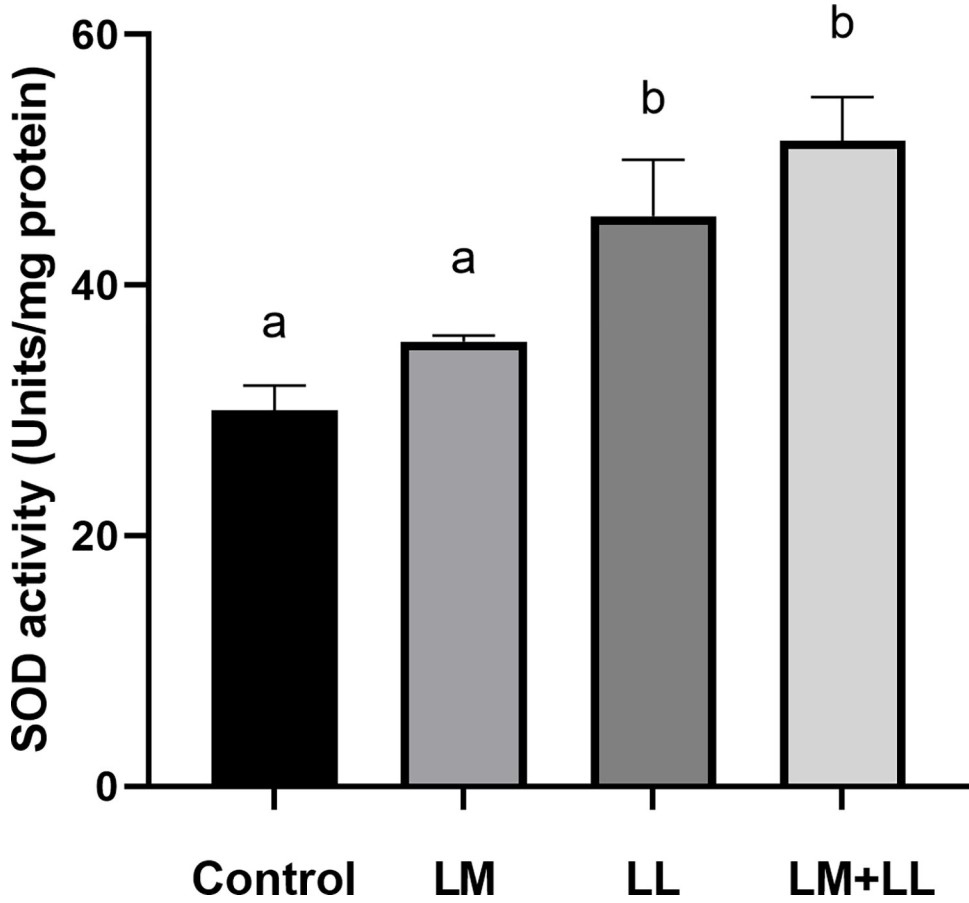

**Fig 6. Superoxide dismutase activity of Nile tilapia fed with control and probiotics supplemented diets.** Data presented as the mean ±SEM from two combined independent experiments (n = 10 fish/group). Bars with different alphabet are statistically different (P ≤ 0.05). Bars with the same alphabet are statistically insignificant different (P ≥ 0.05). * LM; LL—- *L. mesenteroides*; *L. lactis.*

to be influenced by various probiotic species. For instance, in brown trout (*Salmo trutta*), supplementation with *L. mesenteroides* has been reported to affect lysozyme activity [54]. Similarly, in grouper (*Epinephelus coioides*), supplementation with *L. lactis* has been found to modulate lysozyme activity [39]. Consistent with previous findings, in our study, the probiotics *L. mesenteroides* and *L.* lactis have altered the serum lysozyme activity of Nile tilapia. The *L. mesenteroides* and *L. lactis* resulted in increased serum peroxidase values, while the peroxidase content remained unaffected by the mixture diet of *L. mesenteroides* and *L. lactis*, which supports earlier observation in sea bream (*Sparus aurata L*) [27]. Superoxide dismutase facilitates the conversion of highly reactive superoxide radicals into less reactive hydrogen peroxide and plays a crucial role in the primary antioxidant defense mechanisms against oxidative stress. In our study, there was a significant enhancement in superoxide dismutase activities in the experimental group. Comparable findings were reported in *L. rohita*, where dietary supplementation with *Bacillus subtilis* in combination with *Pseudomonas aeruginosa* and *Lactobacillus plantarum*, resulted in similar improvements [34]. Fish possess a distinctive physical barrier comprising epidermal mucus and skin [61], which serves as a component of their nonspecific defense mechanisms, protecting the fish body from direct exposure to environmental stressors and pollutants in the surrounding water. The findings of this study demonstrated an increased

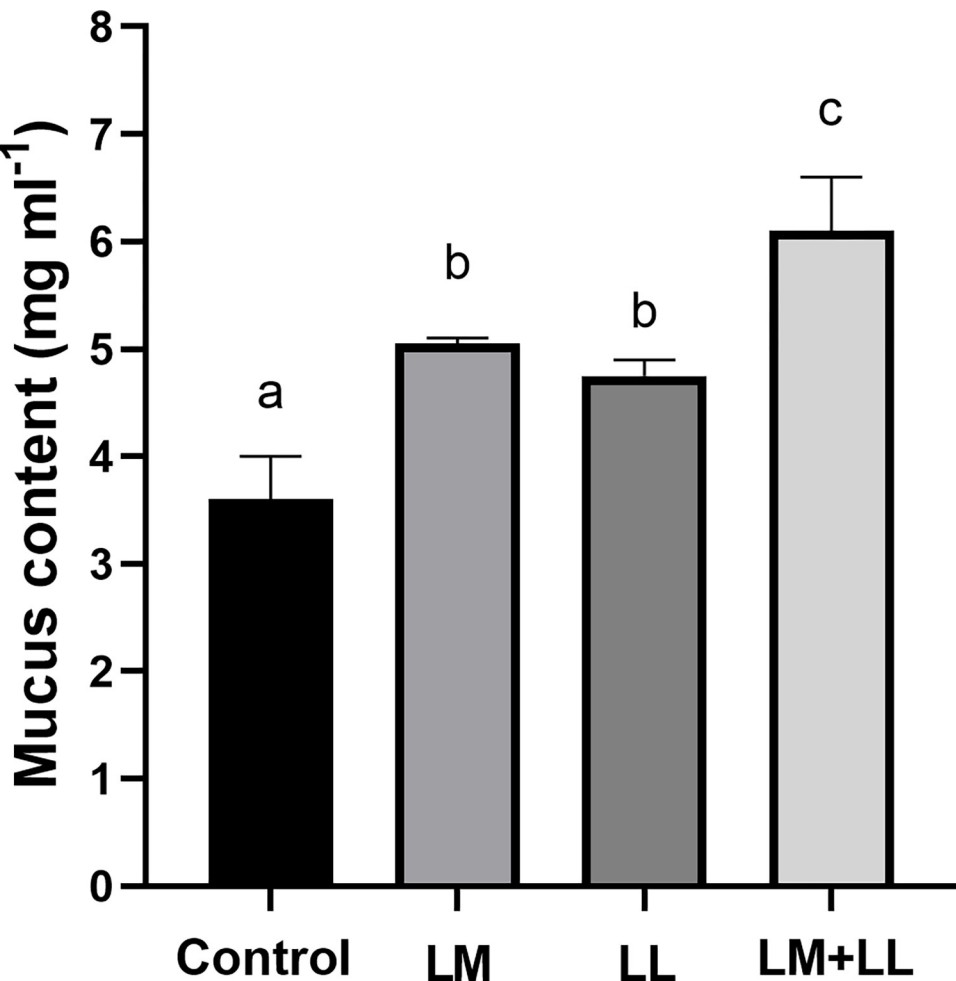

**Fig 7. The effect of probiotics supplemented diets on mucus secretion.** Data presented as the mean ±SEM from two combined independent experiments (n = 10 fish/group). Bars with different alphabet are statistically different (P ≤ 0.05). Bars with the same alphabet are statistically insignificant different (P ≥ 0.05). * LM; LL—- *L. mesenteroides*; *L. lactis*.

secretion of mucus in all fish fed diets supplemented with *L. mesenteroides* or/and *L. lactis* compared to those fed the control diet. Likewise, it has been shown that oral administration of Lactobacillus sp. enhance skin mucus secretion in red sea bream (*Pagrus major*) and gilthead sea bream (*Sparus aurata*) [27,62].

Together current findings showed that the diet supplemented with *L. mesenteroides* and *L. lactis* species exerted more pronounced effects on the innate immune system of Nile tilapia. Consequently, formulations containing multiple probiotic strains were observed to elicit distinct responses in the host immune system compared to formulations containing single probiotic strains. The evidence from this study indicates that in higher vertebrates, well-designed formulations of multistrain or multispecies probiotics have been shown to exhibit health-promoting effects that are lacking in diets containing single-species probiotics [63,64].

The effects of probiotics vary widely and are contingent upon factors such as their sources, dosage, types, and supplementation period. Therefore, when formulating a multi species probiotic mixture, it is essential to thoroughly select candidate strains to optimize their synergistic effects.

The combination of *L. mesenteroides* and *L. lactis* appears to be a suitable mixture in this context. The synergistic effect enabled by the mixture of *L. mesenteroides* and *L. lactis* potentially contributed to the enhanced performance of Nile tilapia. However, further research is needed to confirm this potential.

## Author Contributions

**Conceptualization:** Svetlana Bayantassova, Alexandr Andruchshak, Kaissar Kushaliyev.

**Data curation:** Akylbek Nurgaliyev.

**Formal analysis:** Assel Paritova, Gulbaram Nurgaliyeva, Nurzhan Abekeshev, Altynay Abuova, Faruza Zakirova, Grzegorz Zwierzchowski, Zhaxygali Kuanchaleyev, Saltanat Issabekova, Maigul Kizatova, Zaure Sayakova, Dinara Zhanabayeva, Yelena Kukhar, Ruslan Stozhkov, Botagoz Aitkozhina, Yevgeniy Mayer, Angsar Satbek.

**Investigation:** Akylbek Nurgaliyev, Gulbaram Nurgaliyeva.

**Methodology:** Assel Paritova, Nurzhan Abekeshev, Altynay Abuova, Faruza Zakirova, Grzegorz Zwierzchowski, Zhaxygali Kuanchaleyev, Saltanat Issabekova, Maigul Kizatova, Zaure Sayakova, Dinara Zhanabayeva, Yelena Kukhar, Ruslan Stozhkov, Botagoz Aitkozhina, Yevgeniy Mayer, Angsar Satbek, Alexandr Andruchshak.

**Supervision:** Svetlana Bayantassova, Kaissar Kushaliyev.

**Validation:** Assel Paritova, Akylbek Nurgaliyev, Gulbaram Nurgaliyeva, Nurzhan Abekeshev, Altynay Abuova, Faruza Zakirova, Grzegorz Zwierzchowski, Zhaxygali Kuanchaleyev, Saltanat Issabekova, Maigul Kizatova, Zaure Sayakova, Dinara Zhanabayeva, Yelena Kukhar, Ruslan Stozhkov, Botagoz Aitkozhina, Yevgeniy Mayer, Angsar Satbek, Alexandr Andruchshak.

**Writing – original draft:** Svetlana Bayantassova, Kaissar Kushaliyev.

**Writing – review & editing:** Svetlana Bayantassova, Kaissar Kushaliyev.

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
