## [Decision Letter · Decision Letter 0]

2 Sep 2024

PONE-D-24-34289The dietary effect of multistrain probiotics ( Leuconostoc mesenteroides, Lactococcus lactis ) on growth performance, immune response and gut microbiota in Nile tilapia

(Oreochromis niloticus)PLOS ONE

Dear Dr. Bayantassova,

Thank you for submitting your manuscript to PLOS ONE. After careful consideration, we feel that it has merit but does not fully meet PLOS ONE’s publication criteria as it currently stands. Therefore, we invite you to submit a revised version of the manuscript that addresses the points raised during the review process.

We look forward to receiving your revised manuscript.

Kind regards,

Lee Seong

Academic Editor

PLOS ONE

Journal Requirements:

2. To comply with PLOS ONE submissions requirements, in your Methods section, please provide additional information regarding the experiments involving animals and ensure you have included details on methods of anesthesia and/or analgesia, and efforts to alleviate suffering.

   "This research was funded by the Science Committee of the Ministry of Science and Higher Education of the Republic of Kazakhstan (grant no. AP19576848)"

5. We note that your Data Availability Statement is currently as follows: All relevant data are within the manuscript and its Supporting Information files.

Additional Editor Comments:

Dear author,

After read the manuscript and comments from reviewers, your manuscript needs a major revision

There will be another round of review process

Please response carefully to the comments from reviewers to avoid decline for publication

Please proofread the manuscript before resubmit the revision. Good luck

Reviewers' comments:

Reviewer's Responses to Questions

**Comments to the Author**

1. Is the manuscript technically sound, and do the data support the conclusions?

Reviewer #1: Partly

Reviewer #2: Yes

Reviewer #3: Partly

Reviewer #4: Partly

2. Has the statistical analysis been performed appropriately and rigorously? 

Reviewer #1: No

Reviewer #2: Yes

Reviewer #3: N/A

Reviewer #4: Yes

3. Have the authors made all data underlying the findings in their manuscript fully available?

Reviewer #1: Yes

Reviewer #2: Yes

Reviewer #3: Yes

Reviewer #4: Yes

4. Is the manuscript presented in an intelligible fashion and written in standard English?

Reviewer #1: No

Reviewer #2: Yes

Reviewer #3: Yes

Reviewer #4: Yes

5. Review Comments to the Author

Reviewer #1: Dear authors,

The study is interesting using two LAB as feed additive in nile tilapia farming. However, the major concerns about the paper are as below:

the paper has too many minor errors such scientific name and table format

statistical analysis should conduct HOV and normality tests before anova test

should provide actual p value

should compare to single vs plural LAB analysis before can confirm plural LAB is better

other comments in the attachment

Reviewer #2: Manuscript Number: PONE-D-24-34289

Title: The dietary effect of multistrain probiotics (Leuconostoc mesenteroides, Lactococcuslactis) on growth performance, immune response and gut microbiota in Nile tilapia

Journal: PLOS ONE

This Ms was interesting to isolated new probiotics candidate species isolated from fish and its potential role on growth, antioxidant activity, and immune potential tested in this study in Nile tilapia. But I am not sure of these species done the ability of the probiotics characterization and its in vitro activity is very essential before testing in vivo of aquatic animals. I found some of major mistake of this Ms need to rectify before considering of the article in this journal as seen in below:

Line 22, 46: aquaculture industry growing and developed rapidly not just “two decades” you can change ‘few decades’

Line 22, 47: add ‘animal’ after future

Line 26: add ‘influxes of gut microbiota’ after fish

Line 29,30,34,35,38,40,117,192,193,186,198,211,227,232,234,391,393,401,404,406,410,418,445,461,463,484,466,467,469,480,482,492,493: ‘Leuconostoc mesenteroides and Lactococcus lactis’ change to ‘L. mesenteroides and L. lactis’

Line 36: ‘lactis diet’ change to ‘lactis supplemented diet’

Line 39: ‘lactis groups’ change to ‘lactis supplemented diet groups’

Line 42: keywaord limit to 4 words

Line 54: Include ‘https://doi.org/10.1080/10641260500320845’

Line 55: Include ‘https://doi.org/10.1016/j.aquaculture.2011.03.039’

Line 56: Include ‘https://doi.org/10.1016/j.fsi.2019.04.036; https://doi.org/10.1016/j.fsi.2018.12.051’

Line 57: ‘bacteria’ change to ‘population’

Line 58: ‘have the capability to enhance’ change to ‘were ability to increase’

Line 60: include ‘https://doi.org/10.1111/jam.14628’

Line 61: ‘dietary’ change to ‘nutritional’

Line 66: Include ‘(Oreochromis niloticus)’ after Nile tilapia

Line 67: different fish species, specify?

Line 68: ‘freshwater aquaculture’ change to ‘freshwater fish Nile tilapia’

Line 70: Delete ‘in Nile tilapia’

Line 82: Can check any health or clinical test before acclimatization?

Line 89: control diet means commercial diet or formulated diet?

Line 90: ‘24-hour’ change to ‘24-h’

Line 98: ‘48 h’ change to ’48-h’

Line 109: ’95 ◦C, 95 ◦C, 55 ◦C 72 ◦C 72 ◦C’ change to ‘95◦C, 95◦C, 55◦C 72◦C 72◦C’

Line 114: Commercial feed, trade name, company name, where the diet purchase?

Line 117-125: The subculture of probiotics and how to quantify, not clearly describe..

Line 139: ‘24 h’ change to ’24-h’

Line 154: ‘1h’ change to ‘1-h’

Line 156: ‘10 minutes’ change to ‘10 min’

Line 161: Escherichia coli to be italics

Line 166: space between 10 and μl; Micrococcus lysodeikticus to be italics

Line 175: ‘minutes’ change to ‘min’

Line 193: ‘weight gain (WG), final body weight (FBW), and specific growth rate (SGR)’ change to ‘WG, FBW, and SGR’

Line 195: ‘protein efficiency ratio (PER) and feed efficiency ratio (FER)’ change to ‘PER and FER’

Line 202: in table 2: ‘Nile tilapia (Oreochromis niloticus) fed with Leuconostoc

mesenteroides and Lactococcus lactis’ change to ‘Nile tilapia fed with L.

mesenteroides and L. lactis’

Line 220,230,245,272,293,315,337,359: ‘experimental diets’ change to ‘probiotics supplemented diets’

Line 226: ‘alternative complement pathway (ACP)’ change to ‘ACP’

Line 392,467: Remove ‘(Oreochromis niloticus)’

Line 399: L. lactis to be italics

Line 400: Remove ‘(Paralichthys olivaceus)’

Line 420: Lactobacillus to be italics

Line 439: ‘nutrients [42] [43]’ change to ‘compounds [42,43]’

Line 450: Remove ‘(P. olivaceus)’

Line 470: include the scientific name of the sea bream

Line 474: ‘Labeo rohita’ change to ‘L. rohita’

Line 490: ‘include ‘multi species’ before probiotic mixture

All figure are modify simple and clear for printing.

Reviewer #3: The manuscript entitled " The dietary effect of multistrain probiotics (Leuconostoc mesenteroides, Lactococcus lactis ) on growth performance, immune response and gut microbiota in Nile tilapia " by Assel Paritova et al. aim to evaluate the impact of multiple-strain dietary probiotics Leuconostoc mesenteroides and Lactococcus lactis on the growth performance, immunity, and gut microbiota of Nile tilapia (Oreochromis niloticus).

Based on scientific considerations, this manuscript lacks interesting findings that contribute to the field of fish nutrition/immunology. In general, the manuscript is poorly written and contains major concerns and unclear points that the authors need to address in order to improve the quality of the manuscript and meet acceptable standards for a scientific article. In the present study, the authors fail to provide crucial information. Overall, the writing of the manuscript needs improvement, and it would be beneficial for the authors to seek guidance from expert researchers on how to write a good manuscript.

- This manuscript contains only immune parameters, and lack of data about the parameters related to growth and gut microbiota as mention in the objective.

- The experimental design is poor, I propose that authors should compare each of probiotics and see its combination.

- The data are less, not enough for publication in this high standard journal.

So, I think that this manuscript is not suitable to publish in the journal of Animal Feed Science and Technology, so it must be rejected.

Reviewer #4: Interesting research. However,

1. please make it clearer about "multistrain" definitions, In this study, you applied ONLY two genus. maybe two strain?

2. Why did you apply only two replicates?

3. Please check format of scientific names.

4. Bactericidal activity, in the future. Please select the ones that closer affected on the fish studies, tilapia pathogens.

5. In this study, you cannot state "synergistic" effects because NO individual treatment applied.

6. Please check format of references.

6. PLOS authors have the option to publish the peer review history of their article (what does this mean?). If published, this will include your full peer review and any attached files.

Reviewer #1: No

Reviewer #2: No

Reviewer #3: No

Reviewer #4: **Yes: **Chanagun Chitmanat

---

## [Decision Letter · Decision Letter 1]

8 Oct 2024

PONE-D-24-34289R1The dietary effects of two strain probiotics ( Leuconostoc mesenteroides, Lactococcus lactis) on growth performance, immune response and gut microbiota in Nile tilapia (Oreochromis niloticus)PLOS ONE

Dear Dr. Bayantassova,

Thank you for submitting your manuscript to PLOS ONE. After careful consideration, we feel that it has merit but does not fully meet PLOS ONE’s publication criteria as it currently stands. Therefore, we invite you to submit a revised version of the manuscript that addresses the points raised during the review process.

We look forward to receiving your revised manuscript.

Kind regards,

Lee Seong

Academic Editor

PLOS ONE

Journal Requirements:

Reviewers' comments:

Reviewer's Responses to Questions

**Comments to the Author**

1. If the authors have adequately addressed your comments raised in a previous round of review and you feel that this manuscript is now acceptable for publication, you may indicate that here to bypass the “Comments to the Author” section, enter your conflict of interest statement in the “Confidential to Editor” section, and submit your "Accept" recommendation.

Reviewer #1: All comments have been addressed

Reviewer #2: All comments have been addressed

2. Is the manuscript technically sound, and do the data support the conclusions?

Reviewer #1: Yes

Reviewer #2: Yes

3. Has the statistical analysis been performed appropriately and rigorously? 

Reviewer #1: Yes

Reviewer #2: Yes

4. Have the authors made all data underlying the findings in their manuscript fully available?

Reviewer #1: Yes

Reviewer #2: Yes

5. Is the manuscript presented in an intelligible fashion and written in standard English?

Reviewer #1: Yes

Reviewer #2: Yes

6. Review Comments to the Author

Reviewer #1: Dear author,

Thank you for your work

THe manuscript is look better now

However, there are few concerns as follow:

1. Incubation temperature see line 103

2. format for scientific name and table

other comment in the attachment

Reviewer #2: Manuscript Number: PONE-D-24-34289R1

Title: The dietary effects of two strain probiotics (Leuconostoc mesenteroides, Lactococcus lactis) on growth performance, immune response and gut microbiota in Nile tilapia (Oreochromis niloticus)

Journal: PLOS ONE

The revised Ms was improved well the comments provided by the reviewers. Therefore, this current revision may be consider for publication.

7. PLOS authors have the option to publish the peer review history of their article (what does this mean?). If published, this will include your full peer review and any attached files.

Reviewer #1: No

Reviewer #2: No

---

## [Decision Letter · Decision Letter 2]

10 Oct 2024

The dietary effects of two strain probiotics ( Leuconostoc mesenteroides, Lactococcus lactis) on growth performance, immune response and gut microbiota in Nile tilapia (Oreochromis niloticus)

PONE-D-24-34289R2

Dear Dr. Bayantassova,

We’re pleased to inform you that your manuscript has been judged scientifically suitable for publication and will be formally accepted for publication once it meets all outstanding technical requirements.

Kind regards,

Lee Seong

Academic Editor

PLOS ONE

Additional Editor Comments (optional):

Reviewers' comments:

Reviewer's Responses to Questions

**Comments to the Author**

1. If the authors have adequately addressed your comments raised in a previous round of review and you feel that this manuscript is now acceptable for publication, you may indicate that here to bypass the “Comments to the Author” section, enter your conflict of interest statement in the “Confidential to Editor” section, and submit your "Accept" recommendation.

Reviewer #1: All comments have been addressed

2. Is the manuscript technically sound, and do the data support the conclusions?

Reviewer #1: Yes

3. Has the statistical analysis been performed appropriately and rigorously? 

Reviewer #1: Yes

4. Have the authors made all data underlying the findings in their manuscript fully available?

Reviewer #1: Yes

5. Is the manuscript presented in an intelligible fashion and written in standard English?

Reviewer #1: Yes

6. Review Comments to the Author

Reviewer #1: (No Response)

7. PLOS authors have the option to publish the peer review history of their article (what does this mean?). If published, this will include your full peer review and any attached files.

Reviewer #1: No

---

## [Editor Report · Acceptance letter]

15 Oct 2024

PONE-D-24-34289R2 

PLOS ONE

Dear Dr. Bayantassova, 

I'm pleased to inform you that your manuscript has been deemed suitable for publication in PLOS ONE. Congratulations! Your manuscript is now being handed over to our production team.

Kind regards, 

on behalf of

Dr. Lee Seong 

Academic Editor

PLOS ONE